# An In Vitro Colonic Fermentation Study of the Effects of Human Milk Oligosaccharides on Gut Microbiota and Short-Chain Fatty Acid Production in Infants Aged 0–6 Months

**DOI:** 10.3390/foods13060921

**Published:** 2024-03-18

**Authors:** Menglu Li, Han Lu, Yuling Xue, Yibing Ning, Qingbin Yuan, Huawen Li, Yannan He, Xianxian Jia, Shijie Wang

**Affiliations:** 1College of Food Science and Biology, Hebei University of Science and Technology, Shijiazhuang 050000, China; lml15613182651@163.com (M.L.); jingmaoluhan@126.com (H.L.); lihuawenpumc@126.com (H.L.); tina7088@yeah.net (Y.H.); 2Junlebao Dairy Group Co., Ltd., Shijiazhuang 050000, China; xueyuling@jlbry.com (Y.X.); yibingning@gmail.com (Y.N.); yuanqingbin@jlbry.cn (Q.Y.); 3Jiangsu Junlebao Dairy Co., Xuzhou 221000, China; 4Institute of Basic Medicine, Hebei Medical University, Shijiazhuang 050017, China; hbydjiaxianxian@126.com

**Keywords:** human milk oligosaccharides, infant gut microbiota, in vitro fermentation, *Bifidobacteria*

## Abstract

The impact of five human milk oligosaccharides (HMOs)—2′-fucosyllactose (2FL), 3′-sialyllactose (3SL), 6′-sialyllactose (6SL), lacto-N-tetraose (LNT), and lacto-N-neotetraose (LNnT)—on the gut microbiota and short-chain fatty acid (SCFA) metabolites in infants aged 0–6 months was assessed through in vitro fermentation. Analyses of the influence of different HMOs on the composition and distribution of infant gut microbiota and on SCFA levels were conducted using 16S rRNA sequencing, quantitative real-time PCR (qPCR), and gas chromatography (GC), respectively. The findings indicated the crucial role of the initial microbiota composition in shaping fermentation outcomes. Fermentation maintained the dominant genera species in the intestine but influenced their abundance and distribution. Most of the 10 *Bifidobacteria* strains effectively utilized HMOs or their degradation products, particularly demonstrating proficiency in utilizing 2FL and sialylated HMOs compared to non-fucosylated neutral HMOs. Moreover, our study using *B. infantis*-dominant strains and *B. breve*-dominant strains as inocula revealed varying acetic acid levels produced by *Bifidobacteria* upon HMO degradation. Specifically, the *B. infantis*-dominant strain yielded notably higher acetic acid levels than the *B. breve*-dominant strain (*p* = 0.000), with minimal propionic and butyric acid production observed at fermentation’s conclusion. These findings suggest the potential utilization of HMOs in developing microbiota-targeted foods for infants.

## 1. Introduction

Human milk oligosaccharides (HMOs) are a structurally diverse group of carbohydrates exclusive to human milk. In the composition of human milk’s non-liquid components, HMOs rank third in abundance, following closely behind fat and lactose. These bioactive substances play a critical role in shaping the robust growth patterns and facilitating the various stages of infant development throughout the first few years [1,2,3,4]. Currently, researchers have discovered over 200 molecular structures of HMOs in breast milk. HMOs consist of five monomers: glucose (Glc), galactose (Gal), N-acetylglucosamine (GlcNAc), fucose (Fuc), and N-acetylneuraminic acid (Neu5Ac). Moreover, all HMOs have lactose (Gal-β-1,4-Glc) at their reducing end [4]. In spite of the considerable heterogeneity in their molecular structures, a tripartite classification system exists for HMOs depending on the essence of their central structure, encompassing fucosylated neutral types, non-fucosylated neutrals, and sialylated species [5]. HMOs are primarily resistant to direct digestion and absorption within the human body, making them unavailable for immediate utilization by infants. Nevertheless, HMOs have the ability to reach the colon directly and serve as a source of nourishment for specific microorganisms within the infant’s gut [6,7,8]. A significant observation to make is the relatively diminished extent to which the resident gut microbes effectively utilize and metabolize HMOs [9].

In the initial stages of life, *Bifidobacterium* assume a preeminent position within the infant gut microbiome, exhibiting a strong correlation with substantial shifts in the composition and structure of the intestinal microbial community during this critical period [10,11]. When compared to the adult intestine, the intestinal microbiota of infants exhibits lower diversity and a more unstable microbial structure. Currently, studies suggest that early-life enterotypes can be categorized into four groups, with the key microorganisms being Firmicutes, *Bifidobacterium*, *Bacteroides*, and *Prevotella*. As infants grow, the first two microorganisms replace the latter, and the development process of their intestinal microbiota becomes deterministic and predictable [12]. The establishment of early-life intestinal microbiota is directly linked to health risks at each developmental stage [13,14,15].

In light of the extensive array of benefits it imparts, the World Health Organization (WHO) advocates for a period of exclusive breastfeeding extending from birth through the sixth month of infancy for both the baby’s and mother’s well-being [16,17]. Among these benefits, HMOs play a regulatory role in the infants’ intestinal tract, promoting the growth of *Bifidobacterium*. However, the digestive capabilities of HMOs exhibit strain specificity [18,19]. *Bifidobacterium longum subsp. infantis* (*B. infantis*), *Bifidobacterium bifidum* (*B. bifidum*), *Bifidobacterium longum subsp. longum* (*B. longum*), and others can utilize HMOs in the gut. HMOs undergo specific microbial metabolism in the gut, producing short-chain fatty acids (SCFAs). The intestinal epithelium utilizes SCFAs as an energy source, which in turn supports and optimizes the performance of the intestinal barrier system [20,21].

In vitro simulated fermentation systems of specific prebiotics can be employed to assess the interaction of microbiota in the intestine and the fermentation characteristics of prebiotics [22,23]. Currently, fermentation methods applied to fecal bacteria cultures are classified into two categories: batch cultures and continuous cultures. Batch cultures are widely used because of their convenient operation and the quick assessment of the intestinal microbiota’s ability to utilize specific carbon sources [24]. In this study, the yeast extract–casein hydrolysate–fatty acids medium (YCFA) was used, capable of culturing the majority of the microbiota in the intestine. In a study, fecal samples were harvested from six healthy volunteers and utilized for both YCFA-based microbial cultivation and extensive metagenomic sequencing. The outcome disclosed that, on a mean basis, about 93% of the raw sequencing outputs remained consistent between the native fecal microorganisms and their corresponding cultured counterparts among all six contributors [25].

The primary goal of this study was to quantify the abundance of 10 *Bifidobacteria* species in the infant gut microbiota and evaluate the in vitro fermentation outcomes of the infant gut microbiota where *Bifidobacterium* served as the dominant genus in the presence of specific HMOs. At the outset, we recruited a group comprising 41 infants within their first 6 months of life, all born at full term. Utilizing data from a foundational 16S rRNA sequencing examination, we then proceeded to select samples with a relative *Bifidobacterium* abundance exceeding 60% for further investigation through an in vitro manipulation. The selected HMOs included neutral fucosylated HMO (2′-fucosyllactose, 2FL), sialylated HMOs (3′-sialyllactose, 3SL; 6′-sialyllactose, 6SL), and nonfucosylated neutral HMOs (lacto-N-tetraose, LNT; lacto-N-neotetraose, LNnT). Simultaneously, the selected prebiotics currently added to infant formulae, namely Galactooligosaccharides (GOSs) and Fructooligosaccharides (FOSs), were used as positive controls. The contents of 10 special *Bifidobacteria* species and SCFAs were determined after 24 h of fermentation, providing the possibility of adding HMOs to formula milk powder.

## 2. Materials and Methods

### 2.1. Volunteer Recruitment and Sample Collection

We conducted a local recruitment drive in Shijiazhuang, Hebei, where we enlisted 41 infants who had completed their full gestational term and exhibited excellent health. Each infant was meticulously vetted to ensure compliance with the exhaustive list of eligibility standards: (1) age < 6 months; gestational age 37–42 weeks; gestational singleton; (2) feeding method of breastfeeding or mixed feeding or artificial feeding; (3) no intake of antibiotics or probiotic preparations for 2 weeks before sampling, and absence of constipation, diarrhea, or other intestinal diseases. The present study secured an ethical endorsement from the Ethics Committee at Hebei Medical University, under the auspices of approval number 20211111. During sampling, guardians completed a questionnaire survey and provided informed consent.

Fecal samples from infants were instantly introduced into sterile collection tubes. Then, the samples were transported to the laboratory at 4 °C within 2 h, immediately repackaged, and stored at −80 °C. Firstly, 16S rRNA sequencing technology was used to identify microorganisms in infant feces. Samples with a relative abundance of *Bifidobacterium,* exceeding 60%, were selected for in vitro fermentation experiments. Details of the selected samples are presented in Table 1.

### 2.2. Experimental Design for In Vitro Fermentation

#### 2.2.1. Materials

Five HMOs (2FL, 3SL, 6SL, LNT, and LNnT) were provided by Glycom A/S (Hørsholm, Denmark). GOSs and FOSs were purchased from GALAM (Ma’anit, Israel).

#### 2.2.2. In Vitro Fermentation of Infant Fecal Inocula

The selected fecal samples were thoroughly homogenized, and a 10% (*w*/*v*) fecal bacterial suspension was prepared by precisely weighing 1 g of feces into sterile saline. Following a brief centrifugation at 300 rpm to eliminate impurities, 0.5 mL of the fecal suspension was collected using a sterile syringe. The fecal suspension and HMO were mixed and added to the YCFA medium (Hailu Medical Technology Co., Ltd., Hangzhou, China) to achieve an HMO concentration of 8 mg/mL in the medium. The same procedure for FOSs and GOSs was used as a positive control, while the medium containing only the fecal suspension served as a negative control in the experiment. Subsequently, all the media were incubated anaerobically at 37 °C for 24 h in an incubator. Upon completion of fermentation, the samples were stored at −20 °C.

### 2.3. Fecal Sample DNA Extraction and qPCR

Following the manufacturer’s instructions, microbial community DNA was extracted using the TIANNamp Stool DNA kit (TIANGEN BIOTECH, Beijing, China). The primer sequences were employed for the quantitative testing of 10 types of *Bifidobacteria* species, including *Bifidobacterium*, *Bifidobacterium longum subsp. infantis* (*B. infantis*), *Bifidobacterium bifidum* (*B. bifidum*), *Bifidobacterium longum subsp. longum* (*B. longum*), *Bifidobacterium breve* (*B. breve*), *Bifidobacterium dentium* (*B. dentium*), *Bifidobacterium adolescentis* (*B. adolescentis*), *Bifidobacterium angulatum* (*B. angulatum*), *Bifidobacterium pseudocatenulatum* (*B. pseudocatenulatum*), *Bifidobacterium thermophilum* (*B. thermophilum*), *Bifidobacterium animalis* (*B. animalis*) (Table 2). The qPCR reaction mixture (20 μL) comprised 0.8 μM primers, 10 μL 2×SYBR Green Master Mix, and 2 μL DNA. The amplification program included one cycle at 95 °C for 5 min, followed by 40 cycles of 95 °C for 15 s, 58 °C for 30 s, and 72 °C for extension (Applied Biosystems QuantStudio 3, Waltham, MA, USA). Standard curves were normalized to the number of gene copies for each species. The relative amount of a typical bacterium in a fecal sample was defined as the ratio of that bacterium to *Bifidobacterium*.

### 2.4. SCFA Detection by Gas Chromatography

Quantitative assessment of SCFAs in fecal samples using an external standard calibration method for GC was carried out as has been reported [31]. In brief, a minimum of 0.2 g of feces was placed into a centrifuge tube, brought to a volume of 1 mL with water, and thoroughly homogenized. Then, the pH was adjusted to 2.6–2.8 with 10 mM hydrochloric acid. Upon completion of the processing stage, the samples were subjected to centrifugation at a rotational speed of 12,000 rpm for a duration of 15 min. The supernatant was collected and filtered into the sampling bottle with a water system filter membrane (PES, Tianjin Jinteng Experimental Equipment Co., Ltd., Tianjin, China, 0.22 μm). The supernatant (0.5 μL) was determined using a gas chromatography system (Gas Chromatography- Flame Ionization Detector; Agilent, Santa Clara, CA, USA) equipped with a DB-FFAP (Agilent, Santa Clara, CA, USA) column. Chromatographic conditions included ramping the column temperature to 180 °C at 20 °C/min for 1 min, then to 220 °C at 50 °C/min for 1 min; the split ratio was 10:1, with a flow rate of 2.8 mL/min (Nitrogen 99.99%, Shijiazhuang tongtai technology co., ltd., Shijiazhuang, China).

### 2.5. 16S rRNA Sequencing and Bioinformatic Analysis

Assessment of DNA quality was executed via 1% agarose gel electrophoresis. The variable regions spanning V3 to V4 in the bacterial 16S rRNA gene were amplified by means of PCR primers; these included the forward primer designated as F338 (5′-ACTCCTACGGAGGGGGGGCAG-3′) and the reverse primer R806 (5′-GGACTACHVGGGTWTCTAAT-3′). The PCR was performed by taking 30 ng of DNA samples of acceptable quality and the corresponding primers. The PCR amplification products were purified using Agencourt AMPure XP beads and then dissolved in an elution buffer. The fragment ranges and concentrations of the libraries were measured using an Agilent 2100 Bioanalyzer (Agilent, United States), and sequenced only if they met the qualification criteria. To generate operational taxonomic unit (OTU) sequences, duplicated sequences were removed using QIIME2 software (version 2019.4), and clustering was performed at 100% similarity using the DADA2 method. The resulting unique sequences were collectively identified as OTUs.

### 2.6. Statistical Analysis

Statistical analysis of the gathered data was carried out using SPSS Statistics 17 software. Depending on the distribution characteristics, parametric T-tests were employed for normally distributed data, while non-parametric Mann–Whitney U tests were applied to datasets with non-normal distributions. For comparative analyses of microbial diversity among different groups, either the Wilcoxon signed-rank test was used when dealing with two group comparisons or the Kruskal–Wallis H test was utilized in scenarios involving more than two groups. Furthermore, PERMANOVA (permutational multivariate analysis of variance) was conducted to scrutinize the influence of various grouping factors on sample variability. In our statistical testing framework, a two-tailed test approach was adopted, where P values less than 0.05 (*) were considered indicative of significant differences, and those below 0.01 (**) signified highly significant variations.

## 3. Results

### 3.1. Screening of Samples for In Vitro Fermentation of Infant Feces

Following the analysis of 16S rRNA sequencing data from 41 infant fecal samples, 10 samples with a relative abundance of *Bifidobacterium* > 60% were chosen for subsequent in vitro fermentation experiments. It is evident that *Bifidobacteria* were prevalent and dominant in the infant gut, exhibiting a higher relative abundance in donors 1–10 (Mean ± SD: 80.81 ± 11.00%, 62.13–94.35%) compared to the remaining infant fecal samples (Figure 1A). Quantification of the content of the 10 *Bifidobacteria* species was conducted using the qPCR method, and differences in the microbial structure for donors 1–10 were assessed through PCA analysis. There was a division into two distinct clusters and clear separation among donors 1–10, with *B. breve* dominance in one cluster and *B. infantis* dominance in the other (Figure 1B).

### 3.2. Analysis of Species Composition before and after Fermentation

To explore the impact of HMOs as a substrate on the microbiome of a *Bifidobacteria*-dominated infant fecal inoculum (relative abundance of *Bifidobacterium* >60%), we analyzed species composition through 16S rRNA sequencing before and after fermentation (Figure 2A,B). To delve deeper into the changes in *Bifidobacterium* before and after fermentation, we quantified the contents of the 10 *Bifidobacteria* species (Figure 2C,D). The genus-specific shifts observed can be succinctly characterized as a decline in the relative abundance of potentially harmful bacteria, coupled with an increase in the proportion of beneficial bacterial strains (Table 3). Among the 10 specific *Bifidobacterium* species studied, strain specificity was evident in their utilization of HMOs. *B. infantis* and *B. bifidum* displayed an ability to utilize multiple types of HMOs effectively, with 6SL significantly boosting the growth of both (*p* < 0.05). In contrast, the remaining seven *Bifidobacterium* strains—*B. longum*, *B. dentium*, *B. thermophilum*, *B. animalis*, *B. adolescentis*, *B. angulatum*, and *B. pseudocatenulatum*—were unable to utilize a variety of HMO species. Notably, *B. longum*, *B. pseudocatenulatum*, *B. adolescentis*, and *B. angulatum* exhibit the capacity to utilize sialylated HMOs or their degradation products for enhanced metabolic value. Among these, only *B. breve* exhibited a minimal capacity for HMO degradation (Table 4). Furthermore, the utilization ability of these 10 *Bifidobacteria* species for 2FL and sialylated HMOs was stronger compared to that for neutral non-fucosylated HMOs (Specific findings on microbial abundance and distribution for each HMO are shown in Appendix A).

We employed the PERMANOVA method to conduct a meticulous analysis designed to identify and quantify potential significant differences in the ecological community structure across multiple study groups (Table 5 and Table 6). Upon examining these results, it was revealed that from a macroscopic perspective, with the exception of the 3SL group, the community structures in several other HMO groups underwent substantial shifts following fermentation. Delving deeper into the composition of the targeted 10 *Bifidobacteria* species, only the 6SL group displayed a significant alteration post-fermentation.

### 3.3. Analysis of SCFA Content before and after Fermentation

SCFAs are prominent metabolites resulting from the fermentation process by intestinal microbiota, with acetic acid (AA) being the most abundant among them. Following fermentation, there was a significant increase in AA content (*p* < 0.001), while the contents of propionic acid (PA, *p* = 0.314), butyric acid (BA, *p* = 0.872), and valeric acid (VA, *p* = 0.052) remained essentially unchanged. Overall, before and after fermentation, only the AA content exhibited differences among the groups (*p* < 0.05). Hence, the analysis focused solely on the differences in AA among the groups (Figure 3A). Prior to fermentation, the AA content in the LNnT group was significantly higher compared to the 2FL, sialylated HMO, and FOS groups. Following 24 h of bacterial fermentation, sialylated HMOs produced more AA, with the AA content in the 6SL group significantly higher than that in 2FL, FOS, and GOS groups.

### 3.4. Correlation Analysis between Gut Microbiota and SCFAs

We investigated the correlation between the content of dominant genera and 10 *Bifidobacteria* species in the intestine and SCFAs before and after fermentation (Figure 3B,C). Prior to fermentation, the majority of dominant genera in the gut exhibited negative correlations with four SCFAs (AA, PA, BA, and VA). However, *Bifidobacterium* showed significant positive correlations with AA and BA, and *Lactobacillus* displayed significant positive correlations with BA. After fermentation, the positive correlation between *Bifidobacterium* and PA, as well as *Lactobacillus* and VA, increased. Meanwhile, fermentation altered the association of BA with *Escherichia* and *Streptococcus,* as well as PA with *Enterococcus* and *Bacillus*. Additionally, AA exhibited a shift from a positive to a negative correlation with *Streptococcus*, and from a negative to a positive correlation with *Lactobacillus*. Before fermentation, the majority of the 10 *Bifidobacteria* species presented negative correlations with four SCFAs, while *B. bifidum* and *B. breve* presented positive correlations with SCFAs. Following fermentation, the positive correlation between AA and each *Bifidobacterium* (*B. pseudocatenulatum*, *B. thermophilum*, *B. infantis*, etc.) increased. However, the correlation between *B. breve* and AA changed from positive to negative. Moreover, the correlation between *Bifidobacterium* (*B. bifidum and B. breve*) and SCFAs changed after fermentation.

Following 24 h of anaerobic fermentation of HMOs, significant alterations occur in the microbial composition, which consequently impacts the intricate interplay among these microorganisms. These microbial interactions are fundamentally linked to the stabilization and maintenance of a balanced gut microbiota ecosystem [32]. Before fermentation, *Bifidobacterium* displayed a marked positive correlation with *Streptococcus*. However, following the 24 h anaerobic fermentation process, this relationship turned negative (Appendix B, Figure A2). This shift can be potentially attributed to the inability of *Streptococcus* to utilize HMOs, leading to growth inhibition in its population. In contrast, most species of *Bifidobacterium* were capable of utilizing HMOs as a nutrient source for their proliferation and development, which consequently resulted in all post-fermentation correlations among *Bifidobacterium* being positively aligned.

The majority of the targeted 10 *Bifidobacteria* species exhibited positive correlations after fermentation (Appendix B, Figure A2), suggesting that there is mutualistic growth promotion occurring within the *Bifidobacterium* community. It has been highlighted that extracellular HMO degradation products generated by *B. bifidum* can be shared among other *Bifidobacterium* populations, fostering this cooperative growth environment [33].

### 3.5. Different Types of HMOs Lead to Different Species Composition

The analysis of the 16S rRNA sequencing results revealed a significant difference in the Chao1 index after fermentation among different groups, while the Shannon index showed less variation. This indicates that the fermentation of different HMOs led to distinct total species counts in the microbial community, while the microbial diversity remained relatively consistent (Figure 4A,B). Figure 4A clearly illustrates that the total number of microorganisms after HMO fermentation was higher compared to the Blank, FOS, and GOS groups. Through the application of PCoA coupled with a beta diversity analysis utilizing the unweighted UniFrac distance method, our study revealed pronounced disparities within the microbial community’s constitution and prevalence after subjecting different HMOs to fermentation (Figure 4C).

LEfSe was used to analyze the microbial composition of HMOs and FOSs/GOSs (grouped as OT) after fermentation. The OT group exhibited fewer enriched species, specifically *Flavobacteriales* and *Flavobacteriia*, both belonging to Bacteroidetes. Conversely, the HMO group showed more abundant biomarkers, including 25 species from Actinobacteriota, Firmicutes, Proteobacteria, Bacteroidetes, Verrucomicrobia, etc. (Figure 5A). In Figure 5B, species with a significant difference and an LDA score > 2.0 are depicted. The genus *Citrobacter* manifested a distinct divergence across the different groups, exhibiting a statistically significant enrichment in the subset characterized by the utmost abundance of HMOs. Additionally, the LDA score of *Citrobacter* surpassed that of other taxonomic units, indicating its substantial impact on the observed differences between groups.

## 4. Discussion

The current study aimed at investigating the potential of five different commercial HMOs, including 2FL, 3SL, 6SL, LNT, and LNnT, each characterized by specific structural configurations, to act as the sole carbon sources during the in vitro fermentation by the complex microorganisms within the infant gut ecosystem. In order to assess how distinct HMOs influence the structure and diversity within the infant’s gut microbial community, methodologies such as 16S rRNA sequencing, alongside qPCR, were utilized in our study. The focus was on examining alterations in the dominant genera and the 10 species of *Bifidobacteria*. In addition, changes in the content of SCFAs after fermentation of different HMOs were analyzed and correlated with gut microbiota.

Findings from our 16S rRNA sequencing data indicated that the capacity for assimilating HMOs among the gut bacteria did not rely on the prevalence or domination of specific species. Instead, this ability seemed more closely related to the initial makeup of the resident intestinal microbiome. The species within dominant genera remained unchanged after 24 h of fermentation by HMOs with different structures, yet their quantity and distribution underwent significant alterations. The relative abundance of beneficial bacteria, such as *Bifidobacterium* and *Lactobacillus*, increased, while that of potentially harmful bacteria like *Escherichia* and *Enterococcus* decreased. These findings align with previous studies [34,35,36]. In contrast to the well-documented HMO utilization strategies of *Bifidobacteria*, the molecular mechanisms by which *Lactobacillus* species process HMOs remain relatively uncharted. Prior research has indeed shed some light on this subject through genomic analyses of *Lactobacillus* strains, revealing that they possess a rather constrained capacity for fermenting HMOs [37,38]. Recent research findings have illuminated that the lactose manipulator enzyme in *Lactobacillus casei* plays a pivotal role in the transportation and metabolic processing of core-2 N-acetyllactosamine, a key component found within HMOs [39]. This suggests that *Lactobacillus casei* degrades HMO through different metabolic pathways. Moreover, LEfSe analysis revealed a more robust probiotic function associated with HMOs. Species that significantly differed in the HMO group based on LDA scores included *Bacteroides*. *Bacteroides* are known to employ various strategies for degrading HMOs [40]. Notably, *Akkermansiaceae* emerged among the significantly different species associated with HMOs. This family, belonging to Verrucomicrobia and represented by a single member, *Akkermansia muciniphila* [*AKK*], is recognized as a potential probiotic. Studies propose that *AKK* can thrive on breast milk by utilizing HMOs [41].

This study revealed that HMOs promote the growth of beneficial bacteria, particularly *Bifidobacterium*. Hence, HMOs were initially dubbed the “bifidus factor” in breast milk. Among the 10 specific *Bifidobacterium* species studied, strain specificity was evident in their utilization of HMOs. *B. infantis* and *B. bifidum* displayed an ability to utilize multiple types of HMOs effectively, with 6SL significantly boosting the growth of both (*p* < 0.05). In contrast, the remaining seven *Bifidobacterium* strains—*B. longum*, *B. dentium*, *B. thermophilum*, *B. animalis*, *B. adolescentis*, *B. angulatum*, and *B. pseudocatenulatum*—were unable to utilize a variety of HMO species. Notably, *B. longum*, *B. pseudocatenulatum*, *B. adolescentis*, and *B. angulatum* exhibit the capacity to utilize sialylated HMOs or their degradation products for enhanced metabolic value. Among these, only *B. breve* exhibited a minimal capacity for HMO degradation, differing from previous studies [42].

The complete breakdown of these HMOs, which come in diverse molecular structures, necessitates the intricate interplay of a series of glycoside hydrolases or membrane transport proteins that are present within the infant gut environment. A majority of *Bifidobacterium* strains have evolved to genetically encode specific enzymes for HMO degradation and transporter proteins, enabling them to selectively utilize HMOs with distinct structures. In the infant’s gut, *Bifidobacterium* employs two distinct strategies for HMO utilization: intracellular digestion and extracellular digestive mechanisms. *B. infantis* can transport HMOs into the cytoplasm through various oligosaccharide transport proteins. Subsequently, intracellular glycosyl hydrolases (GHs) play a role in the degradation of most HMOs [43]. This microorganism also demonstrates a unique extracellular tactic in which HMOs are enzymatically cleaved outside the cell by specific glycosidases into simpler sugars, such as monosaccharides or disaccharides. Subsequently, these hydrolyzed compounds are absorbed into the bacterial cell’s cytoplasm to be metabolized further [40]. Conversely, *B. bifidum* and some *B. longum* utilize extracellular GHs for the initial degradation of HMOs. Subsequently, they transport the intermediates, along with specific transport proteins, into the cell for further catabolic utilization [33].

Indeed, there may be a synergistic effect between *Bifidobacteria*. Studies indicate that *B. bifidum SC555* may release fucose and sialic acid into the environment during its growth on short-chain HMOs, potentially cross-feeding other *Bifidobacteria* [44]. Reflecting on our study results, we observed growth in *B. adolescentis*, *B. dentium*, *B. angulatum*, *B. pseudocatenulatum*, *B. thermophilum*, and *B. animalis* within the HMO group. This growth could be attributed to such cross-feeding or other metabolic adaptations, as previous studies have shown that these species either failed to grow or produced only minimal metabolites in response to HMO stimulation [45,46,47]. However, it has been suggested that *B. dentium* possesses the ability to remove terminal fucose from HMOs [48]. Researchers found a smaller number of glycosyl hydrolase genes in the genomes of *Bifidobacterium angulatum JCM 7096(T)* and *Bifidobacterium pseudocatenulatum JCLA3* [49,50]. The *B. thermophilum* genome contains a β-galactosidase gene, a finding that hints at its potential to degrade HMOs since β-galactosidase can participate in their metabolism; however, the exact mechanism remains uncharted [50].

The production of SCFA stands as a crucial physiological process orchestrated by gut microbiota. This process contributes to sustaining a low-pH environment in the intestine and plays a vital role in connecting microbial communities with host immunity [40,51]. Within HMOs, sialylated HMOs exhibited the highest acid production capability, while neutral fucosylated HMOs and FOSs demonstrated comparable acid production levels. Interestingly, GOSs in the positive control generated the least acid, deviating from findings in previous studies [1,52]. Furthermore, our study, employing *B. infantis*-dominant and *B. breve*-dominant strains as inocula, demonstrated varying levels of AA production by *Bifidobacteria* during HMO degradation. Specifically, the *B. infantis*-dominant strain exhibited significantly higher levels of AA production than the *B.breve*-dominant strain (*p* = 0.000) (Figure A2). There is compelling evidence from studies indicating that among children at elevated risk for asthma, a common observation is lower intestinal abundance of AA, which may allude to an underlying association between AA and the progression of allergic pathologies in early life [53]. Martin and colleagues observed a direct correlation between an increase in the abundance of *Bifidobacterium* and a concomitant rise in AA content. This finding suggests that the proliferation of *Bifidobacterium* is positively associated with enhanced AA production within the gut environment [54]. Interestingly, there was minimal production of PA and BA at the conclusion of fermentation. Conversely, in a separate study using *Bacteroides*-dominated inocula, PA and BA were generated in the later phases of fermentation [52]. This discrepancy may stem from distinct HMO utilization strategies employed by *Bifidobacterium* and *Bacteroides*. This observation highlights that distinct initial microbial compositions result in varying metabolites. While the preliminary data suggest a potential link, establishing a definitive correlation between the initial gut microbiota composition and the metabolic fingerprints that emerge following HMO fermentation would require a considerably wider and more inclusive research cohort, thus allowing for more precise insights into this intricate dynamic.

During the ultimate phase of our study, we delved into the interplay between SCFAs and the gut microbiota prior to and following fermentation. The transition in the correlation patterns that occurred is plausibly connected to the reconfiguration of the gut microbiota’s composition and its altered metabolic behavior as a result of the fermentation process [55]. As expected and consistent with prior research [56], the correlation between *Bifidobacteria* and SCFAs strengthened following 24 h of fermentation. Nevertheless, at the species-level of *Bifidobacteria*, the robust negative correlation between microbiota and metabolites was diminished through HMO fermentation. Notably, the correlation of *B. infantis* and *B. adolescentis* with AA was also modified. In conclusion, the inclusion of prebiotics in the diet leads to substantial alterations in the composition and metabolic functionality of the gut microbiota, highlighting their pivotal role in modulating the gut environment. Notably, a separate study demonstrated a significant association between *B. pseudocatenulatum* and *B. longum* with AA in the gut of 6-year-old children, whereas *B. adolescentis* exhibited a significant association with AA in the gut of adults [57]. Interestingly, these findings differ from the results of the present study, suggesting distinct correlations between microbiota and metabolites in the gut among various age groups. Therefore, it is recommended to tailor nutritional supplements according to the specific interplay between distinct microbiota and metabolites, considering the variations across different age groups.

Several limitations characterize this study. Firstly, the modest sample size constrained our ability to draw conclusions from a diverse array of infant subjects. To mitigate potential biases stemming from individual variability, it is essential to expand our research by recruiting a larger population for future studies. Secondly, we recognized that the analysis was confined to a select few microbial metabolites, thus impeding a thorough comprehension of the multifaceted connections between the diverse microbial populations and their associated metabolic outputs. A more holistic approach would involve subjecting both pre- and post-fermentation fluids to extensive metabolomic analysis to uncover hidden patterns and connections.

## 5. Conclusions

In summary, the fermentation of HMOs by the infant gut microbiota did not alter the dominant genera within the microbial population. However, it induced significant changes in the composition and distribution of these microorganisms. Broadly speaking, there was a notable increase in the relative abundance of beneficial bacteria, particularly *Bifidobacterium* species. Among the 10 specific *Bifidobacterium* species studied, strain specificity was evident in their utilization of HMOs. *B. infantis* and *B. bifidum* displayed an ability to utilize multiple types of HMOs effectively, with 6SL significantly boosting the growth of both (*p* < 0.05). In contrast, the remaining *Bifidobacterium* strains were unable to utilize a variety of HMO species. Post-fermentation, acetic acid (AA) content increased significantly (*p* < 0.001), reaching levels approximately between 500 and 1500 μg/g. Conversely, the contents of propionic acid (PA, *p* = 0.314), butyric acid (BA, *p* = 0.872), and valeric acid (VA, *p* = 0.052) remained largely unchanged. Using fecal inocula dominated by either *B. infantis* or *B. breve*, our results demonstrated that different dominant microorganisms led to varying levels of AA production. Specifically, the *B. infantis*-dominant inoculum generated significantly higher amounts of AA compared to the *B. breve*-dominant one (*p* = 0.000). Ultimately, we underscored that the fermentation characteristics of the five HMOs were more closely related to the initial gut microbial composition rather than the degree of dominance of any particular species.

Our findings highlight the crucial importance of the initial microbial composition of fecal donors in shaping fermentation outcomes. This insight paves the way for the development of personalized formulas tailored to infants with specific dominant microbial profiles, allowing for the optimization of nutritional conditions for each infant group. Looking forward, it is envisioned that future advancements will facilitate the production of HMO-enriched formulas that are more aptly aligned with the gut health needs of the majority of infants, regardless of the specific donor origins.

## Figures and Tables

**Figure 1 foods-13-00921-f001:**
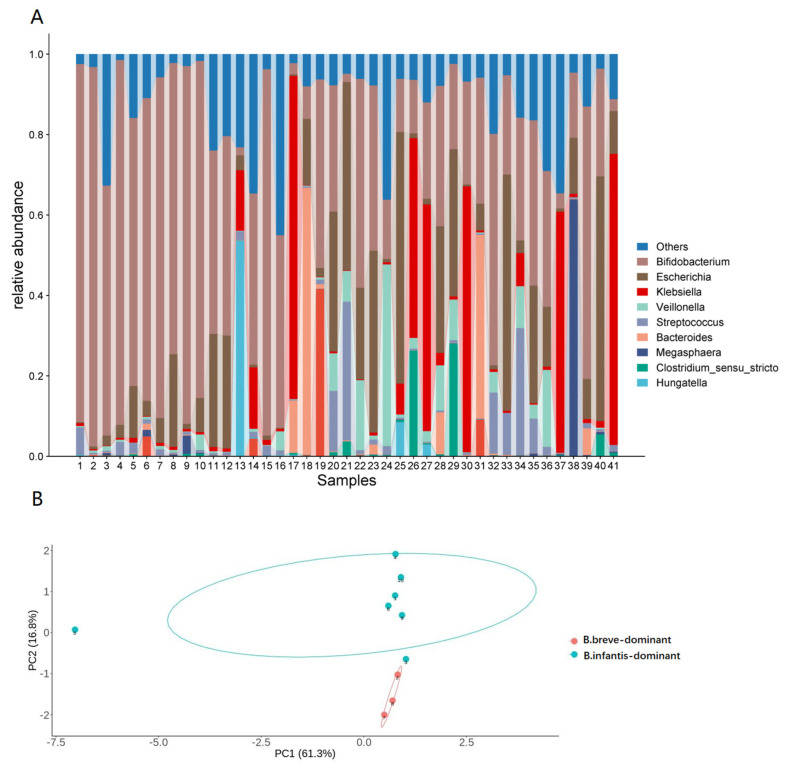
Basic microbial composition of the 41 fecal samples (**A**) and the structural differences in the microbial compositions of donors 1−10 (**B**).

**Figure 2 foods-13-00921-f002:**
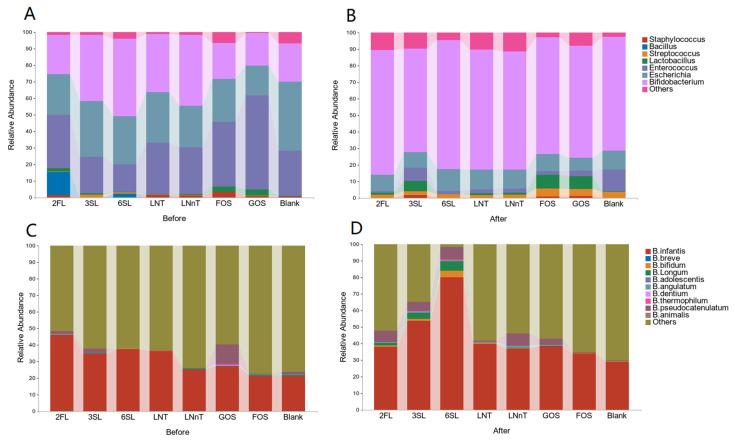
Changes in microbial composition before and after fermentation. (**A**,**B**) The changes of the microbiota in different groups before and after fermentation; (**C**,**D**) The changes in the microbiota of 10 *Bifidobacteria* species in different groups before and after fermentation.

**Figure 3 foods-13-00921-f003:**
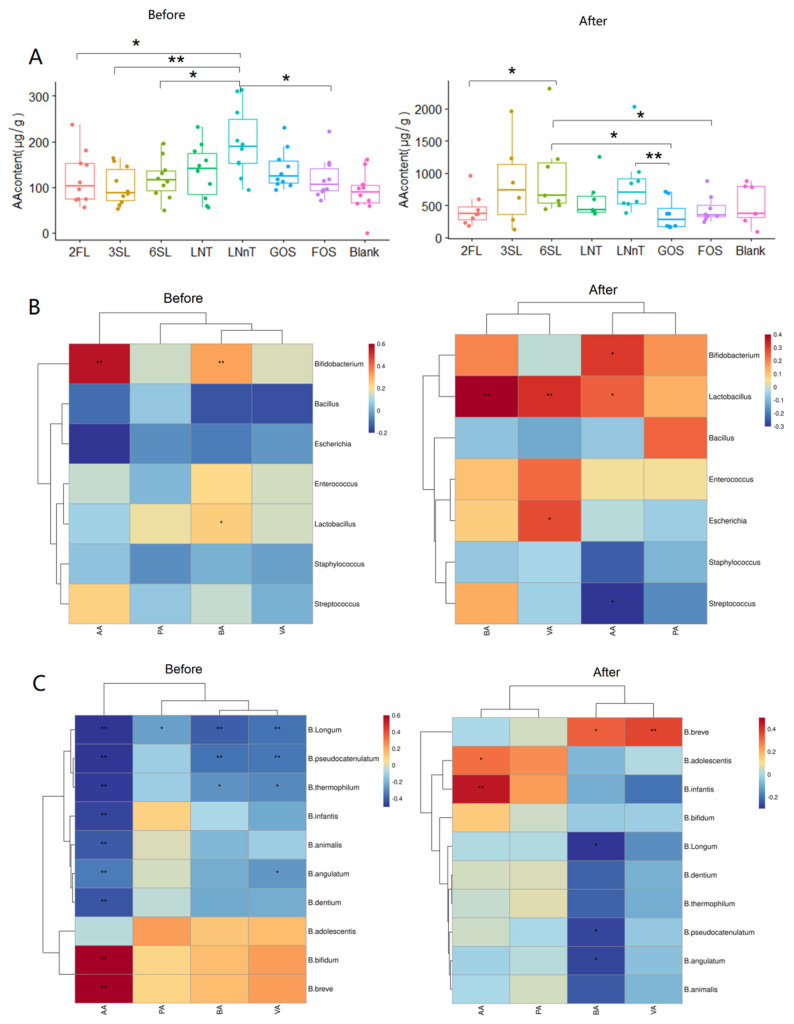
Analysis of SCFA content changes before and after fermentation and correlation with dominant genera and 10 *Bifidobacteria* species. (**A**) Acetic acid content analysis in each group before and after fermentation; (**B**) Correlation analysis of SCFAs and dominant genera before and after fermentation; (**C**) Correlation analysis of SCFAs and 10 *Bifidobacteria* species before and after fermentation. Note: * *p* < 0.05; **: *p* < 0.01.

**Figure 4 foods-13-00921-f004:**
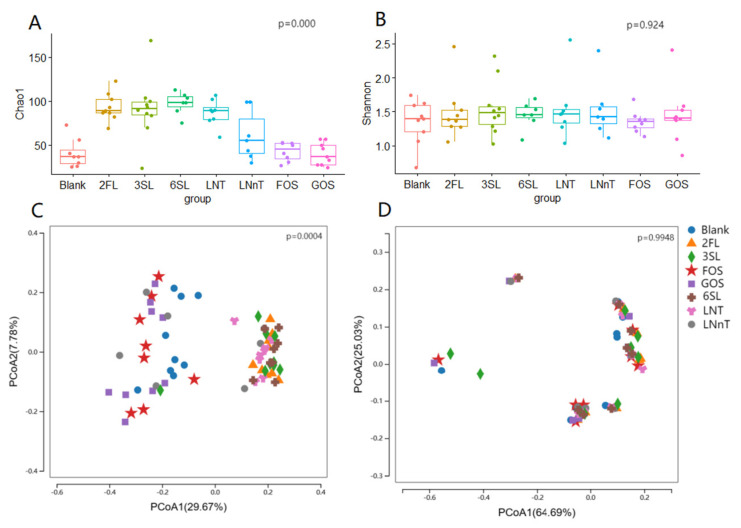
Analysis of microbial α and β diversity after fermentation of different HMOs. (**A**) Analysis of microbial Chao1 index after fermentation of different HMOs; (**B**) Analysis of microbial Shannon index; (**C**) Unweighted UniFrac distance analysis of microbial beta diversity; (**D**) Weighted UniFrac distance analysis of microbial beta diversity.

**Figure 5 foods-13-00921-f005:**
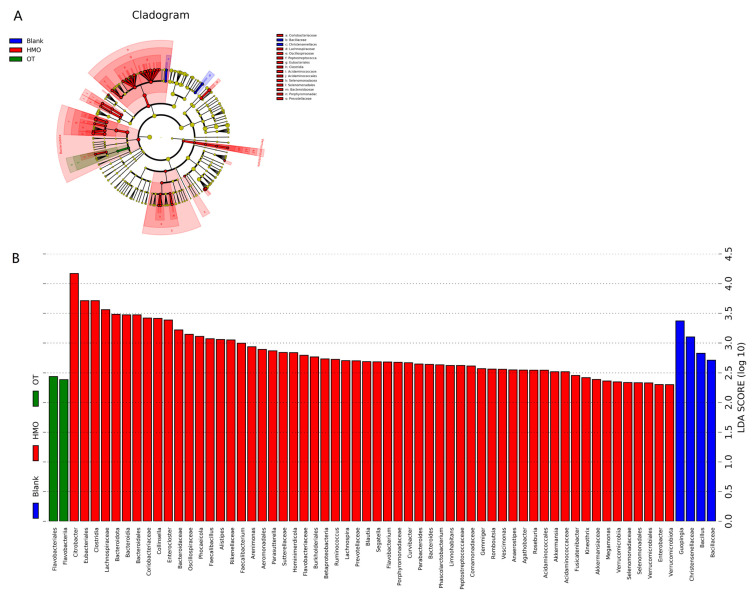
LEfSe analysis of microbial differences between HMOs and OT (FOSs and GOSs) after fermentation. (**A**) LEfSe clustering graph with red nodes representing microorganisms that play an important role in the HMO group; (**B**) LDA graph showing mainly statistically different biomarkers.

**Table 1 foods-13-00921-t001:** Basic information of selected samples.

Number	*Bifidobacterium* Content	Sex	Feeding Mode	Delivery Mode	Age (Month + Day)
1	0.890	Male	Mixed feeding	Vaginal	2 + 20
2	0.944	Female	Exclusive breastfeeding	Vaginal	4
3	0.621	Female	Exclusive breastfeeding	Vaginal	2 + 17
4	0.907	Female	Exclusive breastfeeding	Vaginal	3 + 3
5	0.667	Male	Exclusive breastfeeding	Cesarean	5
6	0.753	Male	Exclusive breastfeeding	Vaginal	3 + 15
7	0.847	Female	Exclusive breastfeeding	Vaginal	3
8	0.724	Female	Exclusive breastfeeding	Vaginal	3 + 17
9	0.890	Female	Exclusive breastfeeding	Vaginal	4 + 12
10	0.838	Female	Exclusive breastfeeding	Vaginal	4

**Table 2 foods-13-00921-t002:** *Bifidobacterium* genus/species PCR-specific primers.

Target	Sequence (5′→3′)	Amplicon Length (bp)	References
*Bifidobacterium*	F-CTCCTGGAAACGGGTGG	549~563	[26]
	R-GGTGTTCTTCCCGATATCTACA		
*B. longum*	F-CGATACCAAGATGGCGTGT	107	[27]
	R-TATGGTTCAGTAGTCACAAAA		
*B. infantis*	F-TTCCAGTTGATCGCATGGTC	823	[28]
	R-GGAAACCCCATCTCTGGGAT		
*B. bifidum*	F-CCACATGATCGCATGTGATTG	278	[28]
	R-CCGAAGGCTTGCTCCCAAA		
*B. breve*	F-CCGGATGCTCCATCACAC	288	[28]
	R-ACAAAGTGCCTTGCTCCCT		
*B. dentium*	F-CCGCCACCCACAGTCT	150	[29]
	R-AGCAAAGGGAAACACCATGTTT		
*B. adolescentis*	F-ATAGTGGACGCGAGCAAGAGA	71	[29]
	R-TTG AAG AGT TTG GCG AAA TCG		
*B. angulatum*	F-TGGTGGTTTGAGAACTGGATAGTG	117	[29]
	R-TCGACGAACAACAATAAACAAAACA		
*B. catenulatum*	F-GTGGACGCGAGCAATGC	67	[29]
	R-AATAGAGCCTGGCGAAATCG		
*B. pseudocatenulatum*	F-AGCCATCGTCAAGGAGCTTATCGCAG	325	[30]
	R-CACGACGTCCTGCTGAGAGCTCAC		
*B. thermophilum*	F-ACTGGTCGCTTCCGCCAAGGATG	326	[30]
	R-CCARGTCAGCMAGGTGRACGATG		
*B. animalis*	F-CACCAATGCGGAAGACCAG	184	[30]
	R-GTTGTTGAGAATCAGCGTGG		

**Table 3 foods-13-00921-t003:** Summary of comparison of differences in key species before and after fermentation.

Genus	After (%)	Before (%)	*p*
*Bifidobacterium*	72.537	29.048	0.0 ***
*Escherichia*	10.643	28.081	0.034 *
*Enterococcus*	1.970	34.812	0.0 ***
*Lactobacillus*	3.451	1.524	0.304
*Streptococcus*	3.117	0.292	0.0 ***
*Bacillus*	0.004	2.021	0.002 **
*Staphylococcus*	0.590	1.288	0.51

Numbers in the third and fourth columns indicate the relative abundance of genera; *: *p* < 0.05; **: *p* < 0.01; ***: *p* < 0.001.

**Table 4 foods-13-00921-t004:** Difference analysis of 10 *Bifidobacteria* species in HMO/FOS/GOS group before and after fermentation.

	2FL	3SL	6SL	LNT	LNnT	FOS	GOS	Blank
*B. infantis*	0.630	0.156	0.018 *	0.817	0.958	1	0.923	0.874
*B. breve*	0.700	0.796	0.655	0.817	0.153	0.386	0.148	0.030 *
*B. bifidum*	0.002 **	0.002 **	0.002 **	0.003 **	0.001 ***	0.001 ***	0.001 ***	0.001 ***
*B. longum*	0.027 *	0.028 *	0.002 **	0.355	0.266	0.211	0.336	0.081
*B. adolescentis*	0.012 *	0.039 *	0.004 **	0.519	0.315	0.043 *	0.178	0.125
*B. angulatum*	0.005 **	0.014 *	0.003 **	0.817	0.039 *	0.043 *	0.124	0.050 *
*B. dentium*	0.043 *	0.071	0.004 **	0.563	0.711	0.211	0.441	0.125
*B. thermophilum*	0.027 *	0.071	0.006 **	0.817	0.153	0.211	0.178	0.153
*B. pseudocatenulatum*	0.021 *	0.039 *	0.003 **	0.728	0.266	0.102	0.441	0.101
*B. animalis*	0.021 *	0.120	0.006 **	0.418	0.081	0.009 **	0.290	0.023 *

The numbers in the table represent whether the changes in the species were significant before and after fermentation in different groups; *: *p* < 0.05; **: *p* < 0.01; ***: *p* < 0.001.

**Table 5 foods-13-00921-t005:** Distribution difference analysis of intestinal dominant bacteria in each group before and after fermentation.

	Df	SumOfSqs	R2 (Effect_Size)	F.Model	Pr (>F) (Adonis_P)
2FL	1	1.119	0.322	7.614	0.001 ***
3SL	1	0.439	0.134	2.628	0.061
6SL	1	0.315	0.203	3.065	0.05 *
LNT	1	0.664	0.279	5.412	0.002 **
LNnT	1	0.427	0.197	3.433	0.013 *
FOS	1	1.210	0.318	7.444	0.003 **
GOS	1	1.561	0.390	10.872	0.001 ***
Blank	1	0.873	0.261	5.294	0.005 **

Df: degree of freedom; R2: Explanation of sample difference by grouping method; *: *p* < 0.05; **: *p* < 0.01; ***: *p* < 0.001.

**Table 6 foods-13-00921-t006:** Distribution difference analysis of 10 *Bifidobacteria* species in each group before and after fermentation.

	Df	SumOfSqs	R2 (Effect_Size)	F.Model	Pr (>F) (Adonis_P)
2FL	1	0.347	0.107	1.800	0.125
3SL	1	0.227	0.085	1.117	0.372
6SL	1	0.596	0.233	3.646	0.001 ***
LNT	1	0.063	0.016	0.216	0.914
LNnT	1	0.317	0.086	1.316	0.237
FOS	1	0.285	0.065	1.042	0.363
GOS	1	0.210	0.048	0.757	0.555
Blank	1	0.237	0.058	0.868	0.421

***: *p* < 0.001.

## Data Availability

The original contributions presented in the study are publicly available. This data can be found at: https://dataview.ncbi.nlm.nih.gov/object/PRJNA985881?reviewer=pkmh3f17o0j0j5mp9vk1qvv88j (accessed on 21 June 2023).

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
