# Peer review of "An In Vitro Colonic Fermentation Study of the Effects of Human Milk Oligosaccharides on Gut Microbiota and Short-Chain Fatty Acid Production in Infants Aged 0–6 Months"

_foods, 2024, doi:10.3390/foods13060921_

Round 1

Reviewer 1 Report

Comments and Suggestions for Authors

I respond to some observations made by the editor of the document "In vitro fermentation results of human milk oligosaccharides by infant intestinal microbiota were related to initial microbial composition"。

This study investigated how oligosaccharides influence the microbiota of infants.

Generally, oligosaccharides are used as prebiotics and added to foods to help probiotics survive.    In this work, it is proposed to know the microbiota possessed by infants derived from delivery mode, which could elucidate the use of prebiotics and to enhance the microbiota present in them.

They are using 2 prebiotics according to the study: GOS and FOS. They are also evaluating whether these substances are prebiotics due to the probable production of short chain fatty acids by chromatography. I consider that the methodology applied is correct and does not need any improvement.

The conclusions are in accordance with the results and suggest how the composition of the microbiota could influence the fermentation of HMOs.

The references are appropriate and well cited.

The tables and figures are in accord with the format. I recommend using scientific names in italics.

In my comments and suggestions I considered that the document has important information about the microbiota present in infants and that this allows it to be a very useful tool for possible conditions or measures that should be considered to avoid diseases in this population.

Reviewer 2 Report

Comments and Suggestions for Authors

In the publication presented here, the authors evaluated the effect of in vitro fermentation of five breast milk oligosaccharides on the intestinal microflora and metabolites of short-chain fatty acids in infants aged 0-6 months.
The article is well written and contains interesting results, but needs minor improvements.
 Specific comments:
- The paper contains editorial errors e.g. lack of spaces before brackets with cited literature items, inconsistencies in italics (e.g. Bacteroides italicised or not);
- A chromatogram of the analysis of short-chain fatty acid metabolites should be included in the supplementary materials;
- Please complete the conclusions and describe the perspectives for future research.

Reviewer 3 Report

Comments and Suggestions for Authors

Reviewer comments:

The manuscript entitled “In vitro fermentation results of human milk oligosaccharides by infant intestinal microbiota were related to initial microbial composition”. The manuscript presents the impact of five human milk oligosaccharides (HMOs) on the gut microbiota and short-chain fatty acid (SCFA) metabolites in infants aged 0-6 months was assessed. However, several points need addressing before proceeding to the next process. The points and suggestions are as follows:

1.       Elaborate on specific findings regarding changes in microbial abundance, distribution, and SCFA levels for each HMO?

2.       Line 153, section 2.516. S rRNA Sequencing, what does it mean S rRNA means? The author means 16S rRNA. Rectify the typographical error.

3.       Sections 3.2 and 3.3 mention the analysis of species composition before and after fermentation. Consider merging these sections or highlighting the unique aspects of each analysis to avoid redundancy.

4.       Sections 3.2 - Provide insights into the potential reasons for the observed changes in specific genera? Are there specific metabolic pathways or interactions that could explain these variations?

5.       Sections 3.2 - The significance of B.infantis dominance and the implications of the observed differences in probiotic function among HMOs, particularly in the 6SL group?

6.       Sections 3.3 - What might be the physiological implications of the increased AA content, especially in the context of infant gut health? Speculate on the potential role of specific microbial groups in AA production?

7.       Sections 3.3 - Discuss the biological significance of the shifts in correlations, especially those involving Bifidobacterium and other dominant genera, before and after fermentation?

8.       Sections 3.4 – Author states that “Bifidobacterium species and SCFAs changed after fermentation”. How might these changes in correlations reflect the metabolic activities of specific Bifidobacterium species during fermentation, and what implications do they have for SCFA production and gut health?

9.       In section 3.4, a correlation analysis between gut microbiota and SCFAs is mentioned. Provide more information on the strength and direction of these correlations?

10.   The manuscript mentions that most of the 10 Bifidobacteria strains effectively utilized HMOs. Provide specific details on the utilization patterns of each HMO by these strains?

11.   In the Conclusions section, the manuscript suggests the potential utilization of HMOs in developing microbiota-targeted foods for infants. Expand on the practical implications of their findings and propose potential future research directions in this context?

12.   Review the manuscript for grammatical clarity and consistency in terminology. Ensure that scientific terms are used accurately and consistently throughout the manuscript.

13.   Address typographical errors throughout the manuscript.

14.   Utilize symbols wherever applicable according to scientific conventions.

15.   Clarify how your research is innovative and contributes to the field.

16.   Discuss any difficulties encountered during the experiments.

17.   Address any limitations of the research and discuss prospects for future work.

18.   Ensure consistency in terminology and formatting throughout manuscript.

Based on the identified points, major revisions are recommended.

Round 2

Reviewer 3 Report

Comments and Suggestions for Authors

The manuscript is improved but the similarity rate is high. The manuscript may be accepted for publication after reducing the similarity.

Author Response

We would like to thank you again for taking the time to review our manuscript. Upon receipt of your feedback, we promptly undertook a meticulous self-examination and comprehensive revision of the entire manuscript. We have carefully revised potentially problematic segments within the text to ensure that all content is original and accurately represented. Furthermore, we employed professional plagiarism detection software to re-evaluate the revised draft. We are steadfast in our belief that through this extensive process of refinement and enhancement, the quality of our paper has been elevated. In this revised version, changes to our manuscript were all highlighted within the document by using red-colored text. Once again, we extend our heartfelt gratitude for your rigorous scrutiny and constructive critique, which have been instrumental in guiding us towards improving the overall integrity of our work.